# What Difference Does it Make? Risk-Taking Behavior in Obesity after a Loss is Associated with Decreased Ventromedial Prefrontal Cortex Activity

**DOI:** 10.3390/jcm8101551

**Published:** 2019-09-27

**Authors:** Trevor Steward, Asier Juaneda-Seguí, Gemma Mestre-Bach, Ignacio Martínez-Zalacaín, Nuria Vilarrasa, Susana Jiménez-Murcia, Jose A Fernández-Formoso, Misericordia Veciana de las Heras, Nuria Custal, Nuria Virgili, Rafael Lopez-Urdiales, Amador García-Ruiz-de-Gordejuela, José M Menchón, Carles Soriano-Mas, Fernando Fernandez-Aranda

**Affiliations:** 1School of Psychological Sciences, University of Melbourne, Parkville, Victoria 3010, Australia; Trevor.steward@unimelb.edu.au (T.S.); jmenchon@bellvitgehospital.cat (J.M.M.); 2Ciber Fisiopatología Obesidad y Nutrición (CIBERObn), Instituto Salud Carlos III, Feixa Llarga s/n, 08907 Barcelona, Spain; Gemma.mestre.bach@gmail.com (G.M.-B.); sjimenez@bellvitgehospital.cat (S.J.-M.); tono.fernandez@ciberisciii.es (J.A.F.-F.); 3Department of Psychiatry, Bellvitge University Hospital-IDIBELL, C/Feixa Llarga s/n, 08907 Barcelona, Spain; ajuaneda@idibell.cat (A.J.-S.); imartinezz@outlook.es (I.M.-Z.); ncustal@bellvitgehospital.cat (N.C.); 4Ciber Salud Mental (CIBERSAM), Instituto Salud Carlos III, Feixa Llarga s/n, 08907 Barcelona, Spain; 5Department of Clinical Sciences, School of Medicine, University of Barcelona, C/Feixa Llarga s/n, 08907 Barcelona, Spain; 6Department of Endocrinology and Nutrition, University Hospital of Bellvitge-IDIBELL, C/Feixa Llarga s/n, 08907 Barcelona, Spain; nuriag@bellvitgehospital.cat (N.V.); mvirgili@bellvitgehospital.cat (N.V.); rafaellopez@bellvitgehospital.cat (R.L.-U.); 7CIBERDEM-CIBER de Diabetes y Enfermedades Metabólicas Asociadas, Instituto de Salud Carlos III, C/Feixa Llarga s/n, 08907 Barcelona, Spain; 8Neurology Department, Bellvitge University Hospital-IDIBELL, C/Feixa Llarga s/n, 08907 Barcelona, Spain; mveciana@bellvitgehospital.cat; 9Bariatric and Metabolic Surgery Unit, Service of General and Gastrointestinal Surgery, Bellvitge University Hospital-IDIBELL, C/Feixa Llarga s/n, 08907 Barcelona, Spain; gordeju@icloud.com; 10Departament of Psychobiology and Methodology in Health Sciences. Universitat Autònoma de Barcelona, 08193 Barcelona, Spain

**Keywords:** obesity, fMRI, risk, reward, sensation seeking, impulsivity

## Abstract

Altered activity in decision-making neural circuitry may underlie the maladaptive food choices found in obesity. Here, we aimed to identify the brain regions purportedly underpinning risk-taking behavior in individuals with obesity. Twenty-three adult women with obesity and twenty-three healthy weight controls completed the Risky Gains Task during functional magnetic resonance imaging (fMRI). This task allows participants to choose between a safe option for a small, guaranteed monetary reward and risky options with larger rewards. fMRI analyses comparing losing trials to winning trials found that participants with obesity presented decreased activity in the left anterior insula in comparison to controls (*p* < 0.05, AlphaSim corrected). Moreover, left insula activation during losses vs. wins was negatively correlated with UPPS-P questionnaire sensation seeking scores. During safe vs. risky trials following a loss, the control group exhibited increased activation in the ventromedial prefrontal cortex (vmPFC) (*p* < 0.05, AlphaSim corrected) in comparison to the OB group. Moreover, vmPFC response in the obesity group during post-loss trials was negatively correlated with risky choices on the task overall. As a whole, our findings support that diminished tuning of the insula towards interoceptive signals may lead to a lack of input to the vmPFC when weighing the costs and benefits of risky choices.

## 1. Introduction

The worldwide number of adult women with obesity has increased from approximately 69 million in 1975 to 390 million in 2016 [1]. The combination of high chronic stress and risk-taking has been found to increase the likelihood of visceral fat gain in women over the course of 18 months, thereby suggesting that impulsivity may be a potentially beneficial intervention target for obesity prevention [2]. Still, several questions regarding risk-taking among individuals with obesity remain unsettled, including whether differences in risk taking are due to an underestimation of the potential drawbacks of a making a risky decision or whether high-risk behaviors are contained only to the realm of food stimuli. 

Risk-taking can be broadly defined as a predisposition to select options with potential for large beneficial or adverse outcomes over alternatives that provide smaller benefits or adverse outcomes [3]. One recent meta-analysis examining performance on the Iowa Gambling Task determined decision-making to be impaired in individuals with obesity, though the clinical importance of non-food-related decision-making deficits remains unclear in terms of the impact such deficits have on weight itself [4]. Other research has found that individuals with obesity have greater difficulty inhibiting dominant behaviors and may be more sensitive to reward than healthy weight controls [5,6], thereby increasing the probability of engaging in risk-taking. Studies using functional magnetic resonance imaging (fMRI) have linked higher adiposity to hypo-responsivity of somatosensory regions during monetary reward feedback, indicating that blunted activation in somatosensory regions may underpin altered reward feedback learning [7].

The Risky Gains Task (RGT) has been utilized during fMRI scanning in diverse populations to examine neural activation patterns while individuals weigh less rewarding, safe choices against more rewarding, risky choices [8]. Research has linked RGT-derived brain activity to the personality trait of sensation seeking, suggesting that, in individuals with high levels of this trait, positive responses to reward outweigh the impact of the equivalent loss [3]. The RGT has consistently been found to elicit activations in the insula [9,10], a region that is crucial for integrating homeostatic signals with external information and expected outcomes [11]. Indeed, one longitudinal study following a sample of adolescents during a weight-loss intervention found that increased activation in the anterior insula during the RGT was positively associated with reductions in the body mass index (BMI) and fat mass [12]. Relatedly, individual differences in interoceptive sensitivity are thought to modulate decision-making processes orchestrated in the ventromedial prefrontal cortex (vmPFC) by guiding one’s awareness of internal cues of hunger or satiety [13]. Obesity and eating disorders characterized by impulsive behaviors have been linked to poor interoceptive sensitivity [14,15], which may lead a failure to integrate interceptive feedback outcomes into the decision-making process [16].

The present study, which was part of a larger national project examining impulsivity in obesity and eating disorders, sought to assess the neural correlates of risk-taking between adult women with obesity and healthy controls (HC) by means of the RGT during fMRI. Additionally, we aimed to determine whether brain-derived activation levels during the RGT are linked to sensation seeking traits. We only included women in our sample given the strong evidence supporting sex differences in sensation seeking traits and risk taking [17] and the fact that previous research has found that the direction of the association between sensitivity to reward and weight is gender-specific [18]. Based on prior studies using the RGT [10,12,19], we hypothesized that women with obesity would exhibit altered activation patterns in the insula during reward vs. punishment trials. We further hypothesized that RGT behavioral measures would be related to neural activity in prefrontal regions related to decision making and reward sensitivity [9,20].

## 2. Methods

### 2.1. Participants

The study featured 23 adult women with obesity who were recruited from the Bariatric and Metabolic Surgery Unit and the Endocrinology and Nutrition Unit at Bellvitge University Hospital (Barcelona, Spain). Participants with obesity were compared to 23 HC women recruited from the same hospital catchment area. The obesity (36.57 ± 9.75) and HC (30.57 ± 10.96) groups did not significantly differ with regards to age (*p* = 0.056). These data are included in Table 1. All participants underwent the Mini-International Neuropsychiatric Interview (M.I.N.I.) with staff psychologists from the Bellvitge University Hospital Department of Psychiatry to screen for the presence of a psychiatric disorder [21]. This study was undertaken as part of a national project including women with eating disorders and obesity. Complete inclusion and exclusion criteria for this study are included in the Appendix A.

The Bellvitge University Hospital Clinical Research Ethics Committee approved the study (PR146/14). Signed informed consent was obtained from all participants. 

### 2.2. Impulsivity Measure (UPPS-P)

This 59-item self-report UPPS-P questionnaire assesses five facets of impulsivity: negative urgency, positive urgency, lack of premeditation, lack of perseverance, and sensation seeking [22]. Individuals are asked to consider acts/incidents during the last six months when rating their behaviors and attitudes. The Spanish adaptation of the UPPS-P has demonstrated good reliability (Cronbach’s α between 0.79 and 0.93) and external validity [23].

### 2.3. Anthropometric Measures

The Tanita BC-420MA was utilized to assess body composition and to calculate BMI. This noninvasive device (Tanita BC-420MA, Tanita Corp. Tokyo, Japan) uses bioelectrical impedance analysis to measure weight and body composition variables [24]. Height was measured via a stadiometer.

### 2.4. fMRI Risky Gains Task 

The RGT has been described in previous studies featuring diverse clinical populations [3,9,25]. During the task, participants completed trials showing the numbers 20, 40, and 80 in increasing order. These numbers represented the amount of monetary reward (cents) that could be added to their total. Prior to scanning, participants were instructed that choosing 20 was always the safest option but that they could choose to wait to receive, or risk losing, 40 cents, or to wait again to receive or risking losing 80 cents. Unbeknownst to the participants, the RGT included predefined outcome frequencies so that the final amount won remained identical irrespective of whether 20, 40, or 80 cents were selected. Visual feedback was provided after each choice, and the cumulative total of the participants winnings was displayed after the completion of each trial to allow for performance monitoring. Participants were paid the same amount for their participation in the study irrespective of task performance. 

The task was made up of 96 trials lasting 3.5 seconds each. Trials were presented in a predetermined randomized order, with 54 trials being “win” trials and 42 “loss” trials. Responses were measured via a button box; responses outside of the designated timeframe resulted in a loss. 

### 2.5. In-Scanner Behavioral Analysis

The frequency of trials counted as “risky” (±40 or ±80) or “safe” (+20), and as “wins” (+20, +40, or +80) vs. “losses” (−40 or −80) were compared between groups. In addition, the frequency of “safe” choices vs. the number of “risky” choices following a loss were compared between groups, as in previous research [9]. 

### 2.6. Imaging Data Acquisition, Pre-processing, and Analysis

Participants were scanned using a 3T Phillips Ingenia system equipped with a thirty-two-channel phased-array head coil. The fMRI sequence used a single-shot gradient-echo echo-planar imaging (EPI, 2D), with a repetition time of 2000 msec, an echo time of 25 msec, and a pulse angle of 90°, in a 24 cm field of view and an 80 × 80 pixel matrix (readout bandwidth at 2040 Hz/pixel) providing isotropic voxel sizes of 3 × 3 × 3 mm with no gap. A total of 40 interleaved sections, parallel to the anterior-posterior commissure line, were acquired for each whole-brain volume for 8 minutes and 18 seconds. A high-resolution T1-weighted anatomical scan was also acquired to facilitate registration of the EPI data into standard space. Specifically, a three-dimensional fast-spoiled gradient, inversion-recovery sequence with 233 contiguous slices (repetition time, 10.43 msec; echo time, 4.8 msec; flip angle, 8°) in a 24 cm field of view, with a 320 × 320 pixel matrix, readout bandwidth at 144 Hz/pixel, and isotropic voxel sizes of 0.75 × 0.75 x 0.75 mm was used. The total scan time was 5 minutes and 4 seconds

#### 2.6.1. fMRI Image Preprocessing 

All functional images were initially preprocessed using the Wavelet Despike procedure within the BrainWavelet Toolbox [26]. This process removes a range of high and low frequency artifacts from a time series by denoising the synchronized signal transients induced by abrupt physical movements. Next, image preprocessing was performed using the statistical parametric mapping software (SPM 12) toolbox running on MATLAB R2017a. For each participant, functional images were realigned to the mean position of the individual time-series scans and the corresponding high-resolution structural T1 image was oriented to the anterior and posterior commissure (AC-PC) line. Functional scans were then co-registered to the T1 image, which was used for non-lineal normalization to standard Montreal Neurological Institute (MNI) space. Normalization parameters were then applied to functional time-series, which were finally smoothed with an 8 mm full width at half maximum (FWHM) kernel. White matter, cerebrospinal fluid (CSF), and global blood oxygen level-dependent (BOLD) time-series, in addition to translation and rotation movement parameters, were also introduced as confounders in a further denoising step performed with the CONN toolbox [27].

#### 2.6.2. First-Level Analyses

The contrasts of interest defined for first-level (single-subject) analysis were: losses vs. wins, safe vs. risky choices, and safe choices vs. risky choices following a loss. Conditions were modelled as the time elapsed from the beginning of the trial to the time of the participants’ response or to the appearance of punishment feedback. The BOLD response at each voxel was convolved using the SPM12 canonical hemodynamic response function and a 128 s high-pass filter.

#### 2.6.3. Second-Level Analyses

Between-group comparisons in task-induced activations were conducted with two-sample t-tests using group (HC vs. OB) as the main factor. Analyses were carried out within regions generated by extracting and conjoining areas from one-sample (OB and HC) activations for each contrast (*p*< 0.001, uncorrected). Derived peak activation differences following significance thresholding were extracted and entered into an SPSS data matrix to assess their relationship with clinical and RGT behavioral measures.

#### 2.6.4. Significance Thresholding

Statistical significance was determined by a combination of voxel-level and cluster-extent thresholds using the AlphaSim algorithm as implemented in the SPM RESTplus V1.2 toolbox [28]. A minimum spatial cluster extent (*ke*) to satisfy a family-wise error (FWE) rate correction of pFWE < 0.05 over contrasts was applied. Input parameters to AlphaSim included a voxel-level probability of *p* < 0.001, a rmm (cluster edge connected) of 3, a FWHM corresponding to the actual smoothing of the data after model estimation, and the mask volume from the contrast of interest. 

### 2.7. Statistical Analyses of Non-Imaging Data

Statistical analyses of non-imaging data (e.g., sociodemographic variables and UPPS-P questionnaire results) were carried out with SPSS 21 (IBM Corp; Armonk, NY). Between-group comparisons were carried out using independent sample t-tests, and linear associations were estimated using Pearson’s correlations. Shapiro–Wilk tests were performed to confirm the normal distribution of the variables of interest. Effect sizes for mean differences were measured using Cohen’s d coefficient (|d| >0.2–0.5 was considered low, |d|>0.5–0.8 moderate, and |d| >0.8 large) [29]. For Pearson’s correlations, p values were used to determine significance and effect sizes were also reported (|r| >0.10–0.24 was considered low, |r| >0.24–0.37 moderate, and |r| >0.37 large; 23). 

## 3. Results

### 3.1. Clinical and Behavioral Data

As expected, the OB group had a significantly higher BMI and body fat percentage than the HC group (Table 1). No significant differences were found between groups in the UPPS-P impulsivity scores. There were no significant differences between groups in the frequency of risky decisions on the RGT (*p* = 0.215) or in the frequency of risky decisions following a loss (*p* = 0.094).

### 3.2. Imaging Data

#### 3.2.1. Losses vs. Wins

One-sample t-tests showed that both groups commonly activated the precuneus, the dorsomedial and dorsolateral prefrontal cortex, and the anterior insula during the losses vs. wins contrast (Appendix A). Group comparisons using independent sample *t*-tests (AlphaSim corrected, pPWE< 0.05) found that participants in the OB group showed decreased activation in the left anterior insula, the angular gyrus, and the posterior visual areas in comparison to HC during the losses vs. wins contrast (Figure 1, Table 2). 

#### 3.2.2. Safe Choices vs. Risky Choices

One-sample *t*-tests showed that both groups commonly activated the anterior insula, the caudate nucleus, and a cluster comprising the bilateral inferior frontal gyrus during the safe vs. risky choices contrast (Appendix A). No significant differences were found between the groups at our selected threshold (AlphaSim corrected, pFWE < 0.05). However, when comparing the safe choices vs. risky choices following a loss, we found that the HC group presented greater activation in the ventromedial prefrontal cortex (vmPFC) than the OB group (Figure 2, Table 2).

### 3.3. Correlations between Brain Activation Patterns and Behavioral Measures

A negative association between th scores on the UPPS-P Sensation Seeking subscale and insula activation during the losses vs. wins contrast was found in the whole sample (*r* = −0.320, *p* = 0.030; Appendix A) and in the OB group (*r* = −0.419, *p* = 0.047; Figure 1). On the other hand, we did not find this association in the HC group (*r* = −0.049, *p* = 0.823; Appendix A). Moreover, vmPFC activation during safe choices vs. risky choices following a loss was negatively correlated with the frequency of risky choices in both the whole sample (*r* = −0.379, *p* = 0.012; Appendix A) and in the OB group (*r* = −0.478, *p* = 0.029; Figure 2). No significant association was found in the HC group (r (22) = −0,141, *p* = 0.531; Appendix A). 

## 4. Discussion

This study addressed the question whether women with obesity show risk-related neural processing differences and yielded two main results. First, participants with obesity showed decreased activation in the left anterior insula in comparison to controls during the losing trials on the RGT. Moreover, the left anterior insula activation was negatively correlated with sensation seeking scores in the OB group and in the whole sample. Secondly, the OB group displayed reduced activation in the vmPFC when making safe choices vs. risky choices following a loss, and vmPFC activation was negatively correlated with the proportion of risky choices during the task. As a whole, these findings substantiate previous findings linking the insula and vmPFC to the processing of risk during decision making [10,12,19,20]. 

In other studies using the RGT in adolescents with excess weight, decreased insular activation was found during the anticipation of risky choices [19] and served as a predictor of a poor response to weight-loss intervention [12]. The insula is strongly associated with incorporating bodily feedback to shape the cognitive-affective processes that determine future decisions. Interoceptive information that reaches the insula is relayed among neurons throughout the insula’s horizontal axis to bring about an integrated representation of body state and to execute the computation of differences between the predicted and received interoceptive signals [30]. Interestingly, we identified a negative correlation between sensation seeking scores and insula activation during losing trials in the obesity group. Recent studies have linked altered insula activation to the voluntary inhibition of risk taking in those with high levels of sensation seeking [31], as well as to blindness to potential losses [3]. In the context of obesity, the combination of failing to properly incorporate interoceptive signals during a loss (i.e., punishment) and of being insensitive to the body’s satiety signals may predispose some individuals with obesity to unhealthy eating patterns. Furthermore, altered insula activation has been associated with emotional regulation impairments in individuals with excess weight, indicating that deficits in affect regulation may contribute to emotional eating behaviors [32]. This warrants further research exploring whether improvements in interoception can encourage eating behaviors that coincide with the body’s caloric needs [33]. 

Although our groups did not differ behaviorally in overall risk-taking behavior on the RGT, it should be noted that previous studies on methamphetamine-dependent individuals [34] and on cocaine use disorders [9] also did not detect differences in the overall risky choices on the RGT in comparison to the controls. Our neuroimaging analysis did discover differences between groups in vmPFC activity during choices following a loss to the extent that participants in the OB group presented decreased vmPFC activation during the safe choices vs. risky choices following a loss in comparison to the controls. Furthermore, vmPFC activity was negatively correlated with the proportion of overall risky choices in the OB group and in the whole sample, though not in the controls. This finding suggests the existence of a high-risk subset of individuals with OB that is characterized by decreased vmPFC activation and higher risk taking. Indeed, one recent study linked vmPFC alterations to emotion regulation deficits in OB [35], and impulsive eating behaviors have been associated with poorer decision making on the Iowa Gambling Task, a task whose performance is understood to be contingent on vmPFC activity [36]. Previous research has posited that the vmPFC, which is generally active in risk-taking tasks, integrates cognitive control and affect signals to attribute value to stimuli, to associate that value with choices, and to determine whether it should be approached or avoided [37]. In conjunction with the dorsal anterior cingulate and the insula, the vmPFC plays a crucial role in signaling danger and could partly provide an explanation for why participants avoided risky choices when vmPFC was activated [38]. This finding indicates that a subset of individuals with obesity may present reduced vmPFC activation during decision making and may be less efficient in integrating immediate outcomes, leading to maladaptive choices. In addition, a recent meta-analytic study using additional validation in an independent dataset found neuroanatomical differences in obesity to be concentrated in the vmPFC [39], and factors such as iron overload in the vmPFC have been found to be linked to deficits in executive functioning in women with obesity, further supporting that the alterations in decision-making circuitry could be a key contributing factor to decision making in eating behaviors [40].

This study has several limitations that should be considered when interpreting its findings. Due to the its cross-sectional design, our study cannot make inferences regarding whether our described neural differences preceded weight gain, were partially caused by it, or whether a third factor contributed to both neural differences and susceptibility to weight gain. Second, our participants were recruited from a hospital setting while being assessed for bariatric surgery. Our sample does not fully represent individuals with obesity in the general population, and it would be beneficial to attempt the presented results in larger, more diverse populations. Third, factors affecting gonadal hormone levels (e.g., use of exogenous estrogen [including oral contraceptive use]) are known to modulate reward response and were not controlled for. Also, no quantitative data on the participants’ eating behaviors were collected and it would be of interest to examine how neural activation patterns during risky decision-making are linked to the eating for purposes of coping and eating for purposes of reward enhancement [41]. Lastly, the RGT was not designed to temporally isolate neural activity related to the decision making versus outcome phases, given that no jitter is present between one phase and the next [9]. This feature could be addressed in future studies.

## 5. Conclusions

This study provides new evidence showing that altered neural activity in the insula and the ventromedial prefrontal cortex during risky decision making are neural correlates of obesity. Our findings lend support to the notion that a subset of individuals with obesity are subjected to reduced activity in the insula and vmPFC when weighing the costs and benefits of rewards and that these alterations drive entrenched maladaptive decisions. Further efforts should be made to determine whether interventions targeting the improvement of inhibitory control abilities are able to significantly improve treatment outcomes [42]. Last, alterations in risk processing hold promise for serving as a marker of future weight loss and warrant further research to support longitudinal studies on whether pretreatment activation patterns could be leveraged as a clinically useful biomarker, as has been the case for multiple psychiatry disorders [43,44]. As a whole, our findings represent a significant contribution to neurobiological models of obesity and indicate a potential mechanism underlying the symptomatology of this condition.

## Figures and Tables

**Figure 1 jcm-08-01551-f001:**
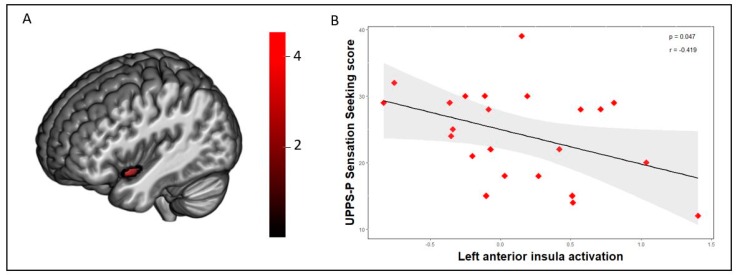
(**a**) Increased activation was found in the left anterior insula in the healthy control group in comparison to the obesity group (AlphaSim voxel level probability = *p* < 0.001, *p* < 0.05 familywise error (FWE) cluster-extent corrected) during the losses vs. wins contrast of the Risky Gains Task. Color bar represents t-values. (**b**) A scatterplot depicting the negative association between extracted activation eigenvalues from the left anterior insula peak (losses vs. wins) and UPPS-P sensation seeking scores in the obesity group [*n* = 23, r (23) =−0.419, *p* = 0.047].

**Figure 2 jcm-08-01551-f002:**
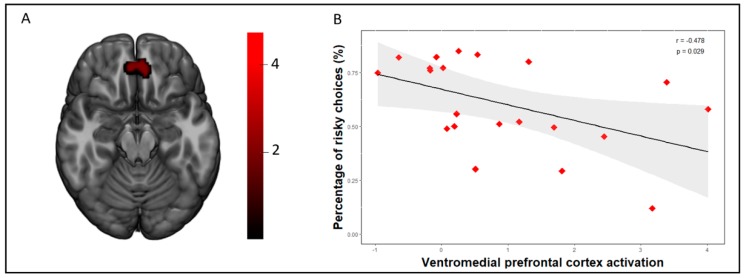
(**a**) Decreased activation was found in the ventromedial prefrontal cortex (vmPFC) in the obesity group compared to the healthy control group (AlphaSim voxel level probability = *p* < 0.001, *p* < 0.05 FWE cluster-extent corrected) on the Risky Gains Task during the safe choices vs. risky choices following a loss contrast. Color bar represents t-values. (**b**) A scatterplot depicting the negative correlation between extracted activation eigenvalues from the vmPFC peak and percentage of total risky choices during the task in the obesity group (*n* = 21, (r (21) = −0.478, *p* = 0.029)).

**Table 1 jcm-08-01551-t001:** Group characteristics.

	Healthy Weight		Obese			
	*n* = 23		*n* = 23			
	Mean	SD	Mean	SD	*p*	|d|
Age	30.57	10.96	36.57	9.75	0.056	0.58
Education (years)	15.80	1.67	14.79	2.29	0.17	0.50
BMI	20.93	1.90	43.35	6.98	<0.001 ^a^	4.39 ^b^
Body fat mass (%)	24.50	5.26	47.11	5.12	<0.001 ^a^	4.35 ^b^
UPPS-P subscales						
Negative urgency	25.22	6.52	28.17	6.39	0.13	0.46
Lack of premeditation	21.78	4.96	23.30	5.14	0.31	0.30
Lack of perseverance	19.30	5.30	22.17	5.16	0.07	0.54
Sensation seeking	28.35	8.44	24.26	6.84	0.08	0.53
Positive urgency	22.52	6.63	20.78	5.13	0.33	0.29
Risky gains task						
Risky choices following loss (%)	59.87	21.05	46.93	29.55	0.09	0.50
Risky choices overall (%)	67.34	16.81	60.26	21.09	0.22	0.37

BMI: Body mass index (kg/m^2^). SD: standard deviation. ^a^ Significant difference (*p* < 0.05). ^b^ Large effect size (|d| > 0.80).

**Table 2 jcm-08-01551-t002:** fMRI Risky Gains Task Results for Group by Outcome Interactions.

		MNI		
		Coordinates		
Contrast	Peak Region	(x,y,z)	Ke ^a^	*t*
Losses vs. wins	Right angular gyrus	46, −54, 30	188	5.78
HC > OB	Right occipital lobe	10, −82, 36	301	4.53
	Left anterior insula	−44, 8, −14	52	4.15
	Left extrastriate visual cortex	−6, −80, 18	101	3.88
Safe choices vs risky choices following a loss	Ventromedial prefrontal cortex	2, 44, −24	175	4.75
HC > OB				

Regions showing between-group differences during the Risky Gains Task (AlphaSim voxel level probability = *p* < 0.001, *p* < 0.05 FWE cluster corrected). MNI: Montreal Neurological Institute. HC: healthy controls. OB: obese. ^a^ Cluster extent in voxels.

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
