# Peer review of "What Difference Does it Make? Risk-Taking Behavior in Obesity after a Loss is Associated with Decreased Ventromedial Prefrontal Cortex Activity"

_jcm, 2019, doi:10.3390/jcm8101551_

Round 1
Reviewer 1 Report
In this study, an experiment was performed with 32 obesity and 23 healthy weight controls, where Risky Gains Task fMRI was performed. Obesity group showed decreased activation in the left anterior insula in comparison to controls during losing trials, and it was shown left anterior insula activation was negatively correlated with sensation seeking scores in the obesity group and in the whole sample. Moreover, obesity group showed decreased activation in the vmPFC in safe choices vs. risky choices following a loss. It was shown that vmPFC activation was negatively correlated with the proportion of risky choices during the task.
The major weakness of this study is inclusion of female subjects only. Is there any specific reason to focus on only women subjects? It should be noted in Introduction and title (if needed).
The findings in this study are very interesting, which can provide insight to understand obesity in neurological aspects. The main conclusions are grounded by convincing results. However, it was difficult to follow the idea regarding the correlation between percentage of risky choices vs. vmPFC activation. First, the same scatter plot or correlation in the whole sample and healthy control is not provided, so cannot appreciate the complete results. Second, what does this result imply? in Table 1 it was shown that there is not significant difference between OB and HC in ‘Risky choices overall’. Doesn’t this contradict with the negative correlation found in OB group? If the negative correlation was found only in OB group, what does it mean? I think it needs further discussion about this.
Here are minor comments.
(Line 38) The acronym UPPS-P is used without introducing it.
(Line 39) During safe vs. risky trials following a loss, the control group exhibited increased activation in the ventromedial prefrontal cortex (vmPFC) => what about obesity group?
(Line 48-55) It seems that this paragraph is added by mistake. This seems the instruction to authors from the journal. Please remove it.
(Line 81-86) I think the main focus of this paragraph is RGT and fMRI. How are researches on vmPFC and the food intake related with RGT? It is confusing. Please restructure the paragraph as needed.
(Line 139) Is this EPI 3D or 2D sequence? Number of slices, scan time, and readout bandwidth are missing. Same for T1w MR image.
(Line 166) Please define post-loss safe choices vs. post-loss risky choices.
(Line 185) Please enumerate the ‘non-imaging data’.
(Line 198) In the table, for Risky choice overall, p should be 0.22 not 0.21 after rounding 0.215.
(Line 234) How was correlation between UPPS-P sensational seeking score vs. Left anterior activation in HC? The correction in the whole sample and OB group is only provided in section 3.3.
(Line 270) Here it is said “… vmPFC activity during choices following a loss to the extent that participants in the OB group 270 presented decreased vmPFC activation during safe choices following a loss in comparison to controls …”. “safe choices” should read “safe choices vs. risky choices” according to Figure 2 and Table 1?
Author Response
Response to reviewers
Reviewer 1:
In this study, an experiment was performed with 32 obesity and 23 healthy weight controls, where Risky Gains Task fMRI was performed. Obesity group showed decreased activation in the left anterior insula in comparison to controls during losing trials, and it was shown left anterior insula activation was negatively correlated with sensation seeking scores in the obesity group and in the whole sample. Moreover, obesity group showed decreased activation in the vmPFC in safe choices vs. risky choices following a loss. It was shown that vmPFC activation was negatively correlated with the proportion of risky choices during the task.
The major weakness of this study is inclusion of female subjects only. Is there any specific reason to focus on only women subjects? It should be noted in Introduction and title (if needed).
Reply: The reviewer brings up a valid point regarding the rationale of only including women in our study sample. Our reasons were threefold: first, there is considerable evidence demonstrating that the directionality of the association between sensitivity to reward and weight is gender-specific and having a sample that includes both men and women runs the risk of not detecting significant effects due to this confounding factor (Dietrich, et al. 2014); second, as mentioned in the introduction, sensation seeking is known to greatly differ between men and women. Including both genders in our sample increases the probability of clinically relevant brain-behavior interactions being confounded by sex; third, this study forms part of a larger national project examining impulsivity in women with obesity and women with eating disorders. Taking into account the low prevalence of certain eating disorders in males, only including females in our sample was deemed necessary in order to allow for greater external validity of our findings. These points are now mentioned in the introduction:
“We only included women in our sample given the strong evidence supporting sex differences in sensation seeking traits and risk taking [16] and the fact that previous research has found that the direction of the association between sensitivity to reward and weight is gender-specific [17].”
The findings in this study are very interesting, which can provide insight to understand obesity in neurological aspects. The main conclusions are grounded by convincing results. However, it was difficult to follow the idea regarding the correlation between percentage of risky choices vs. vmPFC activation. First, the same scatter plot or correlation in the whole sample and healthy control is not provided, so cannot appreciate the complete results. Second, what does this result imply? in Table 1 it was shown that there is not significant difference between OB and HC in ‘Risky choices overall’. Doesn’t this contradict with the negative correlation found in OB group? If the negative correlation was found only in OB group, what does it mean? I think it needs further discussion about this.
Reply: We would like to thank the reviewer for their suggestion on how to improve the manuscript. We have added the requested correlations between risky choices and vmPFC activation to the Supplementary Material. The correlation between vmPFC activation and the percentage of risky choices is only significant in the whole sample and the OB group, suggesting that vmPFC activation and risk taking is a relevant factor in this clinical group. Relatedly, the negative correlation found in the OB group does not contradict the comparison between the OB and HC in “Risky choices overall” since average values do not significantly differ between groups. Instead, this suggests the presence of a high-risk OB subgroup that is characterized by dysfunctional vmPFC activation and higher risk taking. There is significant evidence highlighting the heterogeneity of obesity and this point has been added to the discussion:
“This finding suggests the existence of a high-risk subset of individuals with OB that is characterized by dysfunctional vmPFC activation and higher risk taking. Indeed, one recent study linked vmPFC alterations to emotion regulation deficits in OB [34] and impulsive eating behaviors have been associated with poorer decision making on the Iowa Gambling Task, a task whose performance is understood to be contingent on vmPFC activity [35].”
Here are minor comments.
(Line 38) The acronym UPPS-P is used without introducing it.
Reply: The UPPS-P is in fact not an acronym but is rather reflective of the different subscales that make up the questionnaire. As such, a full name is not available for this questionnaire. See: Lynam, D., Smith, G. T., Cyders, M. A., Fischer, S., & Whiteside, S. A. (2007). The UPPS-P: A multidimensional measure of risk for impulsive behavior.
(Line 39) During safe vs. risky trials following a loss, the control group exhibited increased activation in the ventromedial prefrontal cortex (vmPFC) => what about obesity group?
Reply: We thank the reviewer for bringing this point to our attention. We have modified the abstract to clarify that this significant difference was in comparison to the OB group:
During safe vs. risky trials following a loss, the control group exhibited increased activation in the ventromedial prefrontal cortex (vmPFC) (p<0.05, AlphaSim corrected) in comparison to the OB group.
(Line 48-55) It seems that this paragraph is added by mistake. This seems the instruction to authors from the journal. Please remove it.
Reply: We have removed this section from the journal template.
(Line 81-86) I think the main focus of this paragraph is RGT and fMRI. How are researches on vmPFC and the food intake related with RGT? It is confusing. Please restructure the paragraph as needed.
Reply: We fully agree with the reviewer’s suggestion and this section has undergone significant restructuring:
¨The RGT has consistently been found to elicit activations in the insula [9,10], a region that is crucial for integrating homeostatic signals with external information and expected outcomes [11]. Indeed, one longitudinal study following a sample of adolescents during a weight-loss intervention found that increased activation in the anterior insula during the RGT was positively associated with reductions in body mass index (BMI) and fat mass [12]. Relatedly, individual differences in interoceptive sensitivity are thought to modulate decision-making processes orchestrated in the ventromedial prefrontal cortex (vmPFC) by guiding awareness of internal cues of hunger or satiety [13]. Obesity and eating disorders characterized by impulsive behaviors have been linked to poor interoceptive sensitivity [14,15], which may lead a failure to integrate interceptive feedback outcomes into the decision-making process [16].¨
(Line 139) Is this EPI 3D or 2D sequence? Number of slices, scan time, and readout bandwidth are missing. Same for T1w MR image.
Reply: In accordance with the reviewer’s request, more technical details regarding our neuroimaging sequences have been provided. The fMRI sequence uses echo-planar imaging (EPI) 2D, the number of slices is 40 every 2 seconds, total scan time is 8 minutes and 18 seconds and the readout bandwidth is 2040.7 Hz/pixel. The T1-weighted MR sequences is a 3D sequence, number of slices is 233, total scan time is 5 minutes and 4 seconds and readout bandwidth is 144 Hz/pixel. This information is available in the methods section:
“Participants were scanned using a 3T Phillips Ingenia system equipped with a 32-channel phased-array head coil. The fMRI sequence used a single-shot gradient-echo echo-planar imaging (EPI, 2D), with a repetition time of 2000 msec, an echo time of 25 msec, and a pulse angle of 90°, in a 24-cm field of view and an 80 × 80-pixel matrix, readout bandwidth at 2040 Hz/pixel, providing isotropic voxel sizes of 3 × 3 x 3 mm, with no gap. 40 interleaved sections, parallel to the anterior-posterior commissure line, were acquired for each whole-brain volume for 8 minutes and 18 seconds. A high-resolution T1-weighted anatomical scan was also acquired to facilitate registration of EPI data into standard space. Specifically, a three-dimensional fast-spoiled gradient, inversion-recovery sequence with 233 contiguous slices (repetition time, 10.43 msec; echo time, 4.8 msec; flip angle, 8°) in a 24-cm field of view, with a 320 × 320 pixel matrix, readout bandwidth at 144 Hz/pixel and isotropic voxel sizes of 0.75 × 0.75 x 0.75 mm was used. The total scan time was 5 minutes and 4 seconds.”
(Line 166) Please define post-loss safe choices vs. post-loss risky choices.
Reply: In order to clarify the reviewer’s inquiry, the contrast “post-loss safe choices” vs. “post-loss risky choices” includes trials after a loss that feature either a safe decision or a risky decision. This is now clarified in the manuscript.
(Line 185) Please enumerate the ‘non-imaging data’.
Reply: In agreement with the reviewer’s request, we have given more details on the “non-imaging data”. The non-imaging data includes sociodemographic variables (sex, age, education, body mass index), impulsivity questionnaire results (UPPS-S) and in-scanner behavior (percentage of risk, safe, losses, wins).
“Statistical analyses of non-imaging data (e.g. sociodemographic variables and UPPS-P questionnaire results) were carried out with SPSS 21 (IBM Corp; Armonk, NY).
(Line 198) In the table, for Risky choice overall, p should be 0.22 not 0.21 after rounding 0.215.
Reply: We thank the reviewer for bringing up this error. We corrected the p value in the table.
(Line 234) How was correlation between UPPS-P sensational seeking score vs. Left anterior activation in HC? The correction in the whole sample and OB group is only provided in section 3.3.
Reply: The reviewer is right to inquire regarding the other correlations. The correlation between left anterior insula activation and UPPS-P sensation seeking scores in the HC group was not significant (p-value = 0.82, r = -0.05), meaning that this correlation is only significant in the OB group. We have added this plot to Supplementary Material.
(Line 270) Here it is said “… vmPFC activity during choices following a loss to the extent that participants in the OB group 270 presented decreased vmPFC activation during safe choices following a loss in comparison to controls …”. “safe choices” should read “safe choices vs. risky choices” according to Figure 2 and Table 1?
Reply: We thank the reviewer for identifying this inconsistency. We have corrected this in the manuscript.

Reviewer 2 Report
In the present manuscript, the authors investigate neural activation during a task testing risk-taking behavior in the context of monetary gains and losses. To this end, 23 women with obesity (OB) and 23 healthy controls (HC) underwent functional magnetic resonance imaging (fMRI) while completing the Risky Gains Task (RGT). The authors report less fMRI activation in the left anterior insula in loss compared to win trials in women with obesity compared to HC, and this activation was negatively correlated with a self-report measure of sensation seeking in the whole sample and OB subgroup. In addition, following loss trials, the OB group showed less fMRI activation in the ventromedial prefrontal cortex than the HC group, and this activation was negatively correlated with risky choices in both the whole sample and OB group. No differences in decision-making (e.g., percentage of risky choices) accompanied the neural changes, and whether the relationships observed in the overall sample and OB group also occurred in the HC group remains unclear. The authors discuss their findings as indication for a potential mechanisms underlying obesity.
By examining the neural characteristics of women with obesity during risk-taking behavior, the authors address a timely research topic that could contribute to our understanding of psychological factors that have been identified to play a role in obesity. However, I have several concerns, including a mismatch between hypotheses and analysis, incomplete report of the analysis conducted, and the fact that the current main conclusion and title of the manuscript are not supported by the data, that I would like the author to address before I can recommend the manuscript for publication in the Journal of Clinical Medicine.
Major points:
The hypotheses and analysis of the manuscript do not match. In more detail, the currently stated hypotheses are limited to the neural outcomes of the RGT and their relationship to the behavioral results of the task. Neither the group differences in behavioral outcomes of the RGT nor the use of the UPPS-P are motivated in the Introduction section, and no hypotheses are stated that reveal what the authors predicted to see with these measures in these contexts. As the analysis approach serves to test the outlined hypothesis, the cross-sectional analysis of the behavioral RGT data and analysis of the association between UPPS-P scores and neural activity are sudden and lacks justification. If you had ad hoc hypotheses for those, please include them and provide a clear rationale with them. For example, in the Discussion section, research is cited that links insula activation to sensation seeking. Is this the basis that motivated the authors to include the UPPS-P? Alternatively, please justify why you chose to include the UPPS-P and conduct the analyses that are not linked to your hypotheses. The study involves only female participants (at least in the obese group, and I assume this to be the case in the control group too?). The authors state that they included only women given the sex differences in sensation seeking and risk taking (p. 2, paragraph 5). While I agree that those differences are important to consider, I find myself wondering why the authors did not choose to recruit a mixed sample and include the factor sex in the analysis. Or why did they not test men instead. Thus, I would like to encourage the authors to more clearly explain why women were chosen over men or both. On a related note, reward-related behavior in women has been shown to vary with cycle phase and other aspects impacting gonadal hormones (e.g., estrogen administration). Were the subjects tested in specific cycle phases? Were other factors affecting gonadal hormone levels assessed and/or controlled for (e.g., use of exogenous estrogen [including oral contraceptive use], pregnancy, breastfeeding, untreated thyroid disease)? These questions seem important to exclude alternative explanations for the observed group differences. Correlations between brain activation and UPPS-P are reported for the whole sample and OB group. What about the HC group? It seems important to know the results for the HC group to provide the findings in the OB group with context. In addition, I am curious about other correlations. The UPPS-P has five subscales, and the RGT risky choices can be analyzed overall and following losses and wins, respectively. Similarly, several group differences on the neural level were revealed. How did the authors choose which associations to analyze? Or did they analyze all of them, in which case this needs to be disclosed, the complete results included in the paper, and corrections for multiple comparisons be applied where necessary. The main conclusion and title is not supported by the data. The title refers to “dysfunctional ventromedial prefrontal cortex and insula activation.” However, all the data show is less activation in the stated brain areas in women with obesity compared to HC, while the actual behavior remains unaltered, namely, women with obesity show risk-taking to the same extent as HC in the task used. In my opinion, this does not warrant an interpretation of a lack of functionality. On the contrary, could it not be argued that they show similar performance with less neural activation and thus higher efficiency (I am not saying that this reflects my interpretation of the data but am merely trying to provide a possible alternative account for the observed result pattern). Consequently, can we really judge the meaning of those neural differences? Or is there any meaning/clinical relevance at all? The title and conclusions should be revised accordingly, and this topic should be carefully discussed in the Discussion section. The manuscript refers to previous work using the RGT in individuals with “excess weight”(p. 7, paragraph 2), including fMRI. It seems important to clearly outline how the current study extends the existing literature. For example, the cited study investigated adolescents, while the current study examines adults. Please make those distinctions clear to the readership to allow them to fully understand and appreciate the novelty of your work.
Minor points:
In tasks involving monetary rewards, psychological and neuroeconomic research has shown that the real-life relevance of the task is of essence, namely that participants believe that their choices will translate into factual money at the end of the experiment. From reading the task description, I assume that this was the case here. However, if so, then it would be nice to explicitly state this, as it strengthens the study design. Bioelectrical impedance analysis does not represent the gold standard for the assessment of body composition due to its high variance, and validation of accurate measurement is crucial. Can the authors provide information about the validation procedure for their device? Instead of “p>0.05” for single statistical tests, please report exact p-values. P. 2, paragraph 1: The first paragraph of the Introduction section represents general instructions and needs to be deleted. P. 3, paragraph 1: I believe “with regards to” might need to be changed to “with regard to.” P. 3, paragraph 1: “The obesity and HC groups did not significantly differ with regards to age […] and years of education”: Please provide means and SD for both groups (e.g., by including them in Table 1).
Author Response
Reviewer 2:
In the present manuscript, the authors investigate neural activation during a task testing risk-taking behavior in the context of monetary gains and losses. To this end, 23 women with obesity (OB) and 23 healthy controls (HC) underwent functional magnetic resonance imaging (fMRI) while completing the Risky Gains Task (RGT). The authors report less fMRI activation in the left anterior insula in loss compared to win trials in women with obesity compared to HC, and this activation was negatively correlated with a self-report measure of sensation seeking in the whole sample and OB subgroup. In addition, following loss trials, the OB group showed less fMRI activation in the ventromedial prefrontal cortex than the HC group, and this activation was negatively correlated with risky choices in both the whole sample and OB group. No differences in decision-making (e.g., percentage of risky choices) accompanied the neural changes, and whether the relationships observed in the overall sample and OB group also occurred in the HC group remains unclear. The authors discuss their findings as indication for a potential mechanisms underlying obesity.
By examining the neural characteristics of women with obesity during risk-taking behavior, the authors address a timely research topic that could contribute to our understanding of psychological factors that have been identified to play a role in obesity. However, I have several concerns, including a mismatch between hypotheses and analysis, incomplete report of the analysis conducted, and the fact that the current main conclusion and title of the manuscript are not supported by the data, that I would like the author to address before I can recommend the manuscript for publication in the Journal of Clinical Medicine.
Major points:
The hypotheses and analysis of the manuscript do not match. In more detail, the currently stated hypotheses are limited to the neural outcomes of the RGT and their relationship to the behavioral results of the task. Neither the group differences in behavioral outcomes of the RGT nor the use of the UPPS-P are motivated in the Introduction section, and no hypotheses are stated that reveal what the authors predicted to see with these measures in these contexts. As the analysis approach serves to test the outlined hypothesis, the cross-sectional analysis of the behavioral RGT data and analysis of the association between UPPS-P scores and neural activity are sudden and lacks justification. If you had ad hoc hypotheses for those, please include them and provide a clear rationale with them. For example, in the Discussion section, research is cited that links insula activation to sensation seeking. Is this the basis that motivated the authors to include the UPPS-P? Alternatively, please justify why you chose to include the UPPS-P and conduct the analyses that are not linked to your hypotheses.
Reply: We thank the reviewer for their very useful feedback regarding the manner in which we presented our hypotheses. Though we did allude to sensation seeking traits in the introduction and cite relevant literature on the RGT and sensation seeking, we failed to clearly specify that we aimed to test whether RGT data were linked to UPPS-P sensation seeking scores. This point has been clarified in the introduction:
“Additionally, we aimed to determine whether brain-derived activation levels during the RGT are linked to sensation seeking traits.”
The study involves only female participants (at least in the obese group, and I assume this to be the case in the control group too?). The authors state that they included only women given the sex differences in sensation seeking and risk taking (p. 2, paragraph 5). While I agree that those differences are important to consider, I find myself wondering why the authors did not choose to recruit a mixed sample and include the factor sex in the analysis. Or why did they not test men instead. Thus, I would like to encourage the authors to more clearly explain why women were chosen over men or both.
Reply: As requested, we have explicitly specified in the Methods section that the HC group only included women.
The reviewer brings up a valid point regarding the rationale of only including women in our study sample. Our reasons for not having males in our study sample were threefold: first, there is considerable evidence demonstrating that the directionality of the association between sensitivity to reward and weight is gender-specific and having a sample that includes both men and women runs the risk of not detecting significant effects due to this confounding factor (Dietrich, et al. 2014); second, as mentioned in the introduction, sensation seeking is known to greatly differ between men and women. Including both genders in our sample increases the probability of clinically relevant brain-behavior interactions being confounded by sex; third, this study formed part of a larger national project examining impulsivity in obesity and eating disorders. Taking into account the low prevalence of certain eating disorders in males, only including females in our sample was deemed necessary in order to allow for greater external validity of our findings. These points are now mentioned in the introduction:
“We only included women in our sample given the strong evidence supporting sex differences in sensation seeking traits and risk taking [16] and the fact that previous research has found that the direction of the association between sensitivity to reward and weight is gender-specific [17].”
On a related note, reward-related behavior in women has been shown to vary with cycle phase and other aspects impacting gonadal hormones (e.g., estrogen administration). Were the subjects tested in specific cycle phases? Were other factors affecting gonadal hormone levels assessed and/or controlled for (e.g., use of exogenous estrogen [including oral contraceptive use], pregnancy, breastfeeding, untreated thyroid disease)? These questions seem important to exclude alternative explanations for the observed group differences.
Reply: The reviewer brings up very valid points regarding conducting neuroimaging studies in samples including women. We did not include women who were currently pregnant or breastfeeding in our sample. This point has been specified in our study exclusion criteria listed in the Supplementary Information. However, we did not exclude women taking oral contraceptives and we did not limit scanning women during a specific period of the menstrual cycle. These points are now included in the study limitations section. Our OB sample was recruited from the Bariatric and Metabolic Surgery Unit and the Endocrinology and Nutrition Unit at our public hospital and all patients with a thyroid disease were receiving treatment.
Furthermore, a total of 8 participants (4 from the OB group and 4 from the HC group) had undergone menopause. We reanalyzed our fMRI data excluding this subset of participants and found that our results did not significantly differ. For example, the figure below displays the results of the safe choices vs. risky choices following a loss contrast. The same vmPFC peak was found between groups and T-values were practically identical (4.70 vs. 4.75).
Excluding women with menopause Complete sample
Correlations between brain activation and UPPS-P are reported for the whole sample and OB group. What about the HC group? It seems important to know the results for the HC group to provide the findings in the OB group with context.
Reply: We have added this correlation in the HC group to the Supplementary Material. This correlation was not significant (p-value = 0.82, r = -0.05), therapy suggesting that dysfunctional insula activity and sensation seeking are specifically relevant in the case of obesity.
In addition, I am curious about other correlations. The UPPS-P has five subscales, and the RGT risky choices can be analyzed overall and following losses and wins, respectively. Similarly, several group differences on the neural level were revealed. How did the authors choose which associations to analyze? Or did they analyze all of them, in which case this needs to be disclosed, the complete results included in the paper, and corrections for multiple comparisons be applied where necessary.
Reply: In this study, we only examined associations between sensation seeking and the RGT since this personality trait is most relevant when considering risk-taking behavior in comparison to other UPPS-P subscales such as lack of perseverance (Herman et al., 2018). Furthermore, as stated in the introduction, there is a body of research linking the RGT to sensation seeking in healthy controls and we believe that it would be unwarranted to examine exploratory correlations between other UPPS-P subscales and the RGT in a clinical population without prior validation.
The main conclusion and title is not supported by the data. The title refers to “dysfunctional ventromedial prefrontal cortex and insula activation.” However, all the data show is less activation in the stated brain areas in women with obesity compared to HC, while the actual behavior remains unaltered, namely, women with obesity show risk-taking to the same extent as HC in the task used. In my opinion, this does not warrant an interpretation of a lack of functionality. On the contrary, could it not be argued that they show similar performance with less neural activation and thus higher efficiency (I am not saying that this reflects my interpretation of the data but am merely trying to provide a possible alternative account for the observed result pattern). Consequently, can we really judge the meaning of those neural differences? Or is there any meaning/clinical relevance at all? The title and conclusions should be revised accordingly, and this topic should be carefully discussed in the Discussion section.
Reply: We appreciate the reviewer’s scrutiny of our manuscript title and we agree that it does not properly reflect the findings of the study at hand. As such, it has been changed to read: “What difference does it make? Risk-taking behavior in obesity after a loss is associated with decreased ventromedial prefrontal cortex activity.” This title now properly indicates the nature of the significant differences in activation that were found between groups.
Regarding the reviewer’s other concerns regarding the clinical relevance of our results, we have highlighted in the discussion that our results demonstrate the heterogeneity found in individuals with obesity. For example, there are mixed findings regarding whether individuals with obesity are characterized by increased levels of sensation seeking. However, our findings indicate that those individuals with high levels of sensation seeking present decreased insula activation when facing losses. We believe that our findings could serve as a guide for personalized obesity interventions and provide evidence against one-size-fits-all approaches to weight loss.
“This finding indicates a subset of individuals with obesity may present reduced vmPFC activation during decision making and be less efficient in integrating immediate outcomes, leading to maladaptive choices.”
The manuscript refers to previous work using the RGT in individuals with “excess weight”(p. 7, paragraph 2), including fMRI. It seems important to clearly outline how the current study extends the existing literature. For example, the cited study investigated adolescents, while the current study examines adults. Please make those distinctions clear to the readership to allow them to fully understand and appreciate the novelty of your work.
Reply: We are thankful to the reviewer for bringing up this point. Recent studies using the RGT have examined risky decision making after a loss in clinical populations (e.g. Gowin et al., 2017 & Reske et al., 2015), though no study to date has examined neural activations in individuals with obesity using this contrast. As suggested, we have expanded the contribution of this study to the existing literature in the discussion section.
Minor points:
In tasks involving monetary rewards, psychological and neuroeconomic research has shown that the real-life relevance of the task is of essence, namely that participants believe that their choices will translate into factual money at the end of the experiment. From reading the task description, I assume that this was the case here. However, if so, then it would be nice to explicitly state this, as it strengthens the study design.
Reply: Our Institutional Review Board does not allow compensation amounts to vary between participants. For this reason, we were unable to give factual money to participants reflecting their performance on the RGT. We have clarified this point in the Methods section.
Bioelectrical impedance analysis does not represent the gold standard for the assessment of body composition due to its high variance, and validation of accurate measurement is crucial.
Can the authors provide information about the validation procedure for their device?
Reply: The reviewer is right that other, invasive methods such as dual energy X-ray absorptiometry (DEXA) offer consistently accurate measures of body composition. However, one must consider the tradeoffs of exposing participants to radiation for research purposes when bioelectrical impedance methods have been found to be efficacious for assessing body composition. Details regarding our procedure can be found below:
Body composition was measured by bioelectrical impedance analysis (BIA) (Tanita BC- 420MA, Tanita Corp. Tokyo, Japan). This leg-to-leg body composition analyzer is a simple, accurate, noninvasive and validated method for assessing body composition which provides a small alternating voltage of 90 μA (50/60Hz) via electrodes on metal foot plates. Body weight as well as FM, FFM, MM, TBW, BMR and BMI were calculated for each patient. BC estimates are derived from body fluids making use of proprietary equations based on resistance index, weight, height, age and sex. BIA has been validated against dual-energy X-ray absorptiometry (DEXA) measures and other reference methods.
Browning LM, Dixon AK, Aitken W, Prentice AM, Jebb A. Measuring Abdominal Adipose Tissue : Comparison of Simpler Methods with MRI. Obes Facts. 2011;4: 9–15.
Instead of “p>0.05” for single statistical tests, please report exact p-values.
Reply: The reviewer is right to have us report exact p-values. The p-value for age was 0.056 and the p-value for education was 0.17. This information is in Table 1.
2, paragraph 1: The first paragraph of the Introduction section represents general instructions and needs to be deleted.
Reply: This error was not detected by the editors or the authors and it has been corrected.
3, paragraph 1: I believe “with regards to” might need to be changed to “with regard to.”
Reply: This error has been corrected.
3, paragraph 1: “The obesity and HC groups did not significantly differ with regards to age […] and years of education”: Please provide means and SD for both groups (e.g., by including them in Table 1).
Reply: We have provided additional information regarding this point in the manuscript and it is listed in Table 1.

Round 2
Reviewer 2 Report
I would like to thank the authors for their thoughtful revision. Their replies and edits were very responsive to my comments and nicely addressed my initial concerns.
I only have two final minor suggestions based on the review of the revision that I would like the authors to consider:
The authors mention in their reply that this study was part of a larger national investigation of impulsivity in individuals with obesity and eating disorders and that this – together with the low prevalence of eating disorders in men – was one of the reasons for a solely female sample. It would be great to inform the readers about this broader study context by (1) adding this argument to the other reasons to choose a female sample at the end of the Introduction section, as it now makes the choice for females perfectly justified, and (2) mention the study context in the Methods section. I appreciate that the authors edited the title and agree that it now perfectly reflects the results. Following the same line of reasoning, I would like to kindly ask the authors to change the word “dysfunctional” in the text too, e.g., in Line 332.
Author Response
Reviewer 2
I would like to thank the authors for their thoughtful revision. Their replies and edits were very responsive to my comments and nicely addressed my initial concerns.
I only have two final minor suggestions based on the review of the revision that I would like the authors to consider:
The authors mention in their reply that this study was part of a larger national investigation of impulsivity in individuals with obesity and eating disorders and that this – together with the low prevalence of eating disorders in men – was one of the reasons for a solely female sample. It would be great to inform the readers about this broader study context by (1) adding this argument to the other reasons to choose a female sample at the end of the Introduction section, as it now makes the choice for females perfectly justified, and (2) mention the study context in the Methods section. I appreciate that the authors edited the title and agree that it now perfectly reflects the results. Following the same line of reasoning, I would like to kindly ask the authors to change the word “dysfunctional” in the text too, e.g., in Line 332
Reply (1): We agree with those observation. We now mention in the introduction that this study was part of a larger national project exploring impulsivity in individuals with obesity and eating disorders.
Introduction:
“The present study, which was part of larger national project examining impulsivity in obesity and eating disorders, sought to assess the neural correlates of risk-taking between adult women with obesity and healthy controls (HC) by means of the RGT during fMRI.”
Methods:
“This study was undertaken as part of a national project including women with eating disorders and obesity.”
Reply (2):
In agreement with the reviewer’s suggestion, we have replaced the word “dysfunctional” for “decreased” in the discussion.